# Self-Reported and Parent-Reported School Bullying in Adolescents with High Functioning Autism Spectrum Disorder: The Roles of Autistic Social Impairment, Attention-Deficit/Hyperactivity and Oppositional Defiant Disorder Symptoms

**DOI:** 10.3390/ijerph16071117

**Published:** 2019-03-28

**Authors:** Wen-Jiun Chou, Ray C. Hsiao, Hsing-Chang Ni, Sophie Hsin-Yi Liang, Chiao-Fan Lin, Hsiang-Lin Chan, Yi-Hsuan Hsieh, Liang-Jen Wang, Min-Jing Lee, Huei-Fan Hu, Cheng-Fang Yen

**Affiliations:** 1Department of Child and Adolescent Psychiatry, Chang Gung Memorial Hospital, Kaohsiung Medical Center, Kaohsiung 83301, Taiwan; wjchou@cgmh.org.tw (W.-J.C.); wangliangjen@gmail.com (L.-J.W.); 8035c@cgmh.org.tw (M.-J.L.); 2College of Medicine, Chang Gung University, Taoyuan 33302, Taiwan; alanni0918@yahoo.com.tw (H.-C.N.); sophie.lhy@gmail.com (S.H.-Y.L.); 3Department of Psychiatry and Behavioral Sciences, University of Washington School of Medicine, Seattle, WA 98105-0371, USA; rhsiao@u.washington.edu; 4Department of Psychiatry, Children’s Hospital and Regional Medical Center, Seattle, WA 98105-0371, USA; 5Department of Child and Adolescent Psychiatry, Chang Gung Memorial Hospital, Linkou Medical Center, Taoyuan city 33305, Taiwan; water519@cgmh.org.tw (C.-F.L.); 8802038@cgmh.org.tw (H.-L.C.); 8802026@cgmh.org.tw (Y.-H.H.); 6Department of Psychiatry, Tainan Municipal Hospital (Managed by Show Chwan Medical Care Corporation), Tainan 701, Taiwan; 7Department of Psychiatry, Kaohsiung Medical University Hospital, Kaohsiung 807, Taiwan; 8Department of Psychiatry, School of Medicine, Kaohsiung Medical University, Kaohsiung 807, Taiwan; 9Graduate Institute of Medicine, College of Medicine, Kaohsiung Medical University, Kaohsiung 807, Taiwan

**Keywords:** bullying, autism spectrum disorder, attention-deficit/hyperactivity disorder, oppositional defiant disorder, social impairment

## Abstract

The aim of this study was to examine the prevalence of self-reported and parent-reported bullying victimization, perpetration, and victimization-perpetration and the associations of autistic social impairment and attention-deficit/hyperactivity disorder (ADHD) and oppositional defiant disorder (ODD) symptoms with bullying involvement in adolescents with high functioning autism spectrum disorder (ASD). A total of 219 adolescents with high functioning ASD participated in this study. The associations of sociodemographic characteristics, parent-reported autistic social impairment, and parent-reported ADHD and ODD symptoms with self-reported and parent-reported bullying victimization, perpetration, and victimization-perpetration were examined using logistic regression analysis. The results found that the agreement between self-reported and parent-reported bullying involvement was low. Compared with bullying involvement experiences reported by adolescents themselves, parents reported higher rates of pure bullying victimization (23.7% vs. 17.8%) and victimization-perpetration (28.8% vs. 9.1%) but a lower rate of pure bullying perpetration (5.9% vs. 9.1%). Deficit in socio-communication increases the risk of being pure victims and victim-perpetrators. Parent-reported victim-perpetrators had more severe ODD symptoms than did parent-reported pure victims.

## 1. Introduction

Children and adolescents with autism spectrum disorder (ASD) have been identified as a group vulnerable to school bullying. A recent study found that 19.2% of children with ASD were often victims, versus 4.6% for community children, 10.3% were often involved in teasing others, compared to 2.1% for the community control children [1]. A systematic literature search on children and adolescents with ASD found that the pooled prevalence estimate for school bullying perpetration, victimization and both was 10%, 44%, and 16%, respectively [2]. Bullying victimization was significantly associated with higher risks of depression [3], anxiety [4], suicide [5], poor quality of life [6], and poor educational outcomes [7] in children and adolescents with ASD. The results of previous studies indicated that bullying involvement in children and adolescents with ASD warrants mental health and educational professionals’ attention. Determining the risk factors of involving in school bullying is the essential step to develop programs of prevention, early detection and effective intervention for children and adolescents with ASD.

The risk factors of bullying involvement for children and adolescents with ASD most often examined in previous studies included poor socio-communicative skills and comorbid attention-deficit/hyperactivity disorder (ADHD) and oppositional defiant disorder (ODD). Research has found that poor socio-communicative skills were correlated to peer victimization in children and adolescents with ASD [8,9,10]. Moreover, compared with those with ASD but no ADHD, children and adolescents with ASD and ADHD had increased odds of being bullying victims [10,11] and perpetrators [10,12]. Comorbid ODD also increased the potential for bullying others [11,12]. A recent study even found that the significant association between ASD and being perpetrators or victim-perpetrators disappeared after controlling for comorbid hyperactivity, attention problems, aggression, and conduct problems, while the risk of being bullied in ASD adolescents continued to be significantly elevated [1]. The results of the studies described above indicated that poor socio-communicative skills and comorbid ADHD and ODD have a determining role in bullying involvement among children and adolescents with ASD.

Several important concerns regarding the roles of poor socio-communicative skills and comorbid ADHD and ODD symptoms in the risk of bullying involvement among adolescents with ASD warrant further study. First, the results of previous studies on the ability of adolescents with ASD to recognize bullying involvement are mixed. While some studies have found that adolescents with ASD may misinterpret bullying situations as non-bullying [13,14], other studies have found that adolescents with ASD demonstrate similar levels of ability to identify bullying situations [15,16]. Multiple sources of reports, such as from parents, teachers, and peers, are considered important in identifying bullying involvement among adolescents with ASD. However, whether the rates and related factors of bullying involvement in adolescents with ASD are different between self-report and parent-report warrants further study.

Second, the fifth edition of the Diagnostic and Statistical Manual of Mental Disorders (DSM-5) characterizes ASD in two behavioral domains, including difficulties in social communication and social interaction, and unusually restricted, repetitive behaviors and interests [17]. Social communication deficits may be the results of deficits in theory of mind (ToM) performance among the individuals with ASD [18,19]. The individuals with deficits in ToM performance may have difficulties in understanding others’ mental states and in predicting others’ behaviors resulting from their mental states [20]. These difficulties may compromise social interactions of ASD children and adolescents with peers, make them look odd, and increase the chance of being the targets of bullying. Social communication deficits can make an individual with ASD misinterpret others’ intentions and have inappropriate reactions, which may be interpreted by their peers as provocative actions and become the flashpoint of bullying [21]. Social communication deficits may also limit the ability of the individuals with ASD to build friendships with others and, therefore, reduce the possibility to receive protection and assistance from others when they are bullied. However, ASD has the “spectrum” nature and is composed of subgroups with various symptom severity, cognition, and biological mechanisms [22]. For example, the social responsiveness scale (SRS), an instrument for assessing autistic traits of the individuals with ASD, is composed of various dimensions of autistic social impairment [23,24]. It warrants further study to examine whether various dimensions of autistic social impairment have various relationships with bullying involvement experiences in adolescents with ASD.

Third, research has revealed that bullying victim-perpetrators, defined as people who bully others but are also bullied themselves, are a distinct group and the most troubled amongst all children and adolescents involved in peer bullying [25,26]. Children with ASD identified as bullying victim-perpetrators were more likely to have ADHD, ODD, or conduct disorders [11]. However, whether there are differences in autistic social impairment, ADHD and ODD symptoms between adolescent victim-perpetrators, pure victims and pure perpetrators with ASD warrants further study.

A higher risk of bullying victimization and a lower risk of bullying perpetration were found in children with ASD and without intellectual disability (ID) but not in those with ASD and with ID [1]. Children with a high level of autistic traits were the most likely to involve in bullying [23]. Moreover, children with ASD in full inclusion classrooms were more likely to be victimized than those who spend the majority of their time in special education settings [27]. Therefore, the present study examined the roles of autistic social impairment and comorbid ADHD and ODD symptoms in self-reported and parent-reported bullying involvement in a group of adolescents with high functioning ASD, defined as having full-scale intelligence quotient determined using the Chinese version of the Wechsler Intelligence Scale for Children, fourth edition [28] >80, having verbal communication ability, and currently studying in inclusive classroom but not being pulled out to special education room. We hypothesized that there are differences in the rates and risk factors between self-reported and parent-reported school bullying involvement. Given that communication problems [8], fewer friendships [14], stereotyped behavior and interests [19], and aggressive behaviors [29] have been found to account for the high risk of bullying victimization in adolescents with ASD, we hypothesize that various dimensions of autistic social impairment have various relationships with bullying involvement experiences in adolescents with high functioning ASD. Moreover, given that research on general population found that victim-perpetrators are the most troubled amongst all children and adolescents involved in peer bullying [25,26], we hypothesized that adolescent victim-perpetrators with high functioning ASD have more severe autistic social impairment, ADHD and ODD symptoms between than did pure victims, and pure perpetrators with ASD.

## 2. Methods

### 2.1. Participants

The study participants were enrolled from five child psychiatry outpatient clinics in Taiwan, including three university-affiliated teaching hospitals, one regional teaching hospital, and one child psychiatry specialized clinic. The Taiwan National Health Insurance allows patients visiting the outpatient clinics of teaching hospitals without transference of general practitioners. Therefore, the adolescents of these five child psychiatry outpatient clinics in the present study are representative of those of similar age in Taiwan. The participants were required to meet the following criteria for inclusion in the study: (1) age, 11–18 years; (2) having a diagnosis of ASD according to the DSM-5; (3) full-scale intelligence quotient determined using the Chinese version of the WISC-IV >80; (4) having verbal communication ability; and (5) currently studying in inclusive classroom but not being pulled out to special education room. Those who fitted the criteria were consecutively recruited into this study between August 2013 and July 2016. Parents who had intellectual disability, schizophrenia, bipolar disorder, or any cognitive deficits that resulted in significant community difficulties were excluded. A total of 228 adolescent-parent dyads were invited into this study. Of them, 219 (96.1%) adolescent-parent dyads (219 adolescents with high-functioning ASD, 175 mothers, 33 fathers, and 11 other relatives) agreed to participate in this study and were interviewed by the research assistants based on the research questionnaire. The Institutional Review Board (IRB) of Kaohsiung Medical University (KMUHIRB-20120084).

The systematic review study found that over 50% of children and adolescents with ASD were involved in bullying [2]. The present study focused on adolescents with high functioning ASD, and, therefore, the prevalence of bullying involvement was hypothesized to be 35–40%. Based on a small effect size in logistic regression (odds ratio: 1.0–1.5) with an alpha level of 0.05, a power of 0.80, and the hypothesized rate of bullying involvement 35–40%, 200 participants were deemed to be sufficient to test the hypothesis [30]. The sample of 219 participants was thus determined as adequate.

### 2.2. Measures

#### 2.2.1. Chinese version of the School Bullying Experience Questionnaire (C-SBEQ)

The self-reported and parent-reported C-SBEQ was used to evaluate adolescents’ involvement in school bullying in the previous one year with 16 items answered on a Likert four-point scale range with 0 indicating “never”, 1 indicating “just a little”, 2 indicating “often”, and 3 indicating “all the time” [31,32]. This scale was composed of four four-item subscales evaluating being a victim of passive bullying (items 1 to 4, including social exclusion, being called a mean nickname, and being spoken ill of), being a victim of active bullying (items 5 to 8, including being beaten up, being forced to do work, and having money, school supplies, and snacks taken away), being a perpetrator of passive bullying (items 9 to 12), and being a perpetrator of active bullying (items 13 to 16). Participants who answered 2 or 3 on any item among items 1 to 4, items 5 to 8, items 9 to 12, and items 13 to 16 were identified as self-reported and parent-reported victims of passive bullying, victims of active bullying, perpetrators of passive bullying, and perpetrators of active bullying, respectively. Four groups were also distinguished by type of bullying involvement: pure perpetrators, those who bullied others but were not bullied by others, pure victims, those who were bullied by others but did not bully others, victim-perpetrators, those who were bullied by others and also bullied others, and a neutral group, those who neither bullied others nor were bullied by others. The results of the previous study examining the psychometrics of the C-SBEQ have been described elsewhere and supported that the C-SBEQ has good reliability and validity [32]. In the present study, we invited both adolescents and their parents to rate adolescents’ involvement in school bullying.

#### 2.2.2. Chinese Social Responsiveness Scale (SRS)

The parent-reported Chinese version of the SRS contains 60 items evaluated on a four-point Likert scale that assess adolescents’ extent of autistic social impairment [23,24]. The Chinese version of the SRS was composed of four subscales, including socio-communication, autism mannerisms, social awareness, and social emotion. A higher total score of the subscale indicates greater autistic social impairment. Research has found that the SRS effectively distinguishes between children and adolescents with and without ASD [23,24].

#### 2.2.3. Short Version of the Swanson, Nolan, and Pelham Version IV Scale-Chinese Version (SNAP-IV)

The short version of the SNAP-IV-Chinese version contains 26 items comprising the core DSM-IV-derived ADHD subscales of inattention, hyperactivity/impulsivity, and ODD symptoms [33,34]. Each item is rated on a four-point Likert scale from 0 (not at all) to 3 (very much). Higher total scores of the subscales indicate greater ADHD and ODD symptoms. The Cronbach’s α values of the inattention, hyperactivity/impulsivity, and ODD subscales in the present study were 0.91, 0.91 and 0.92, respectively.

#### 2.2.4. Demographic Characteristics

The present study examined adolescents’ age and sex and parental marital status of (married and living together vs. divorced or separated) and educational duration.

### 2.3. Procedure

Adolescents with high-functioning ASD and their parents were invited to complete research questionnaires. Research assistants conducted interviews to collect adolescents’ self-reported school bullying experiences based on the C-SBEQ. It took five minutes on average for the adolescents to complete the interview. The parents spent 30 min on average to complete the pencil-and-paper C-SBEQ, the Chinese SRS, short version of SNAP-IV and the questionnaire for demographic characteristics. The parents could ask the research assistants if they had problems in completing the questionnaires. Data analysis was performed using SPSS 20.0 statistical software (SPSS Inc., Chicago, IL, USA).

### 2.4. Statistical Analysis

Participants’ demographic characteristics, the levels of social communication deficits and ADHD and ODD symptoms, and the prevalence of self-reported and parent-reported bullying involvement were calculated using descriptive statistics. The associations of demographic characteristics, autistic social impairment, and ADHD and ODD symptoms (independent variables) with being pure bullying victims, pure bullying perpetrators, and victim-perpetrators (dependent variables) were first examined using bivariate logistic regression analysis by using the neutral group as the reference. The significant variables were further selected into multi-variable logistic regression analysis to adjust the effects of other variables. The related factors of being victim-perpetrators were also examined by using pure victims and pure perpetrators as the references. Odds ratio (OR) and its 95% confidence interval (CI) were used to represent the statistical significance.

## 3. Results

### 3.1. Prevalence of School Bullying Involvement

Table 1 shows demographic characteristics and the levels of autistic social impairment and ADHD and ODD symptoms among 219 adolescents with high-functioning ASD. Table 2 shows the rates of self-reported and parent-reported bullying involvement, including being pure victims, pure perpetrators, and victim-perpetrators. Compared with the rates of bullying involvement experiences reported by adolescents themselves, parents reported higher rates of adolescents’ pure bullying victimization (23.7% vs. 17.8%) and victimization-perpetration (28.8% vs. 9.1%) but a lower rate of pure bullying perpetration (5.9% vs. 9.1%). The agreement between self-reported and parent-reported bullying involvement was low (kappa = 0.101). Only 10 (4.6%), 1 (0.5%), and 8 (3.7%) participants were simultaneous self-reported and parent-reported pure victims, pure perpetrators, and victims-perpetrators, respectively. Parents identified a half (*n* = 70) of the self-reported neutrals as pure victims (*n* = 31), pure perpetrators (*n* = 10), or victims-perpetrators (*n* = 29). Parents also identified 38.5% (*n* = 15) of the self-reported pure victims and 55% (*n* = 11) of the self-reported pure perpetrators as victim-perpetrators. Moreover, parents identified 30.8% (*n* = 12) of the self-reported pure victims, 15% (*n* = 3) of the self-reported pure perpetrators, and 30% (*n* = 6) of the self-reported victim-perpetrators as the neutrals.

### 3.2. Correlates of Bullying Involvement

Table 3 shows the results of bivariate logistic regression analysis examining the correlates of self-reported bullying involvement. The results indicated that self-reported pure bullying victims had a shorter maternal education duration and more severe deficit in socio-communication compared with self-reported neutrals. Self-reported pure bullying perpetrators had more severe deficit in socio-communication, inattention and hyperactivity/impulsivity symptoms than did self-reported neutrals. Self-reported victim-perpetrators had more severe deficits in socio-communication, autism mannerism, and social emotion than did self-reported neutrals. No difference in demographic characteristics, social communication deficits, or ADHD and ODD symptoms was found between self-reported victim-perpetrators and self-reported pure victims or between self-reported victim-perpetrators and self-reported pure perpetrators.

Table 4 shows the results of bivariate logistic regression analysis examining the correlates of parent-reported bullying involvement. The results indicated that parent-reported pure bullying victims had more severe deficits in socio-communication, autism mannerism, and social emotion, and inattention and hyperactivity/impulsivity symptoms than did parent-reported neutrals. Parent-reported pure bullying perpetrators had more severe deficits in socio-communication, autism mannerism, and social emotion, and hyperactivity/impulsivity and ODD symptoms than did parent-reported neutrals. Parent-reported victim-perpetrators were older and had more severe deficits in socio-communication, autism mannerism, social awareness, and social emotion, and inattention, hyperactivity/impulsivity and ODD symptoms than did parent-reported neutrals. Parent-reported victim-perpetrators had more severe deficits in socio-communication and autism mannerism, and ODD symptoms than did parent-reported pure bullying victims. No difference in demographic characteristics, autistic social impairment, or ADHD and ODD symptoms was found between parent-reported victim-perpetrators and parent-reported pure perpetrators.

The significant variables were further selected into multivariable logistic regression analysis (Table 5). Both self-reported pure victims and self-reported victim-perpetrators had more severe deficit in socio-communication than self-reported neutrals. Moreover, parent-reported victim-perpetrators also had more severe deficit in socio-communication than did parent-reported neutrals. Parent-reported pure victims had more severe hyperactivity/impulsivity symptoms than did parent-reported neutrals. Parent-reported victim-perpetrators had more severe ODD symptoms than did parent-reported neutrals and parent-reported pure victims. Self-reported pure bullying victims had a shorter maternal education duration than did self-reported neutral group. Parent-reported victim-perpetrators were older than parent-reported neutral group.

## 4. Discussion

The present study found that the agreement between self-reported and parent-reported bullying involvement of adolescents with ASD was low. Several possible etiologies may account for the result. Adolescents may not agree with the perspective of adults on whether some behaviors should be regarded as bullying [35]. ASD may aggravate disagreement of bullying involvement between adolescents and parents. Adolescents with ASD may misinterpret bullying situations as non-bullying [13,14]. Furthermore, the classmates and teachers of adolescents with ASD may interpret adolescents’ autistic behaviors at school as bullying behaviors and report them to their parents, whereas the adolescents with ASD may deny that they have intent on bullying others. Adolescents with ASD and their families may also experience difficulties in interacting and communicating with each other. Such difficulties may further hinder parents from detecting adolescents’ involvement in bullying. Given the crucial role of parents in the prevention of and intervention in bullying involvement, the results of the present study indicate that enhancing parents’ knowledge of ASD adolescents’ bullying involvement at school is an essential step to address bullying. Moreover, mental health and educational professionals must take the self-reported and parent-reported bullying involvement into consideration simultaneously and should not rely on sole information when intervening bullying involvement of adolescents with ASD.

The present study also found that there was a difference in the roles of autistic social impairment and ADHD and ODD symptoms between adolescents with ASD who were involved in self-reported and parent-reported bullying. The deficit in socio-communication was significantly associated with self-reported but not parent-reported pure victimization. Hyperactivity/impulsivity and ODD symptoms were associated with parent-reported but not self-reported pure victimization and victimization-perpetration, respectively. In the present study, autistic social impairment and ADHD and ODD symptoms were rated by parents. Both the data of bullying involvement and related factors from the same reporters may result in shared method variance and influence the significance of the association between parent-reported bullying involvement and deficit in socio-communication, hyperactivity/impulsivity and ODD symptoms. Previous studies using parent-report have a similar significant association of deficit in socio-communicative skills, ADHD and ODD with bullying involvement in children and adolescents with ASD [8,9,10,11,12]. Moreover, although the present study did not assess parental mental health statuses, parents’ psychiatric disorders may influence their observation and report on their offspring’s experiences of bullying involvement. For example, depression may reduce parents’ ability to detect what their offspring encounter in school. Research found that mother’s depression even increased the risk of bullying in their sons [36]. It is also possible that parental ASD, ADHD and ODD may influence parents’ interpretation of social interaction in adolescents. For example, impulsivity may limit the ability of the parents with ASD, ADHD or ODD to comprehensively understand adolescents’ social conflicts and attribute them as bullying or not bullying involvement. The results of the present study indicated that the sources of information may influence the relationships of autistic social impairment, ADHD and ODD symptoms with bullying involvement in adolescents with ASD.

The present study found that self-reported pure victims and victim-perpetrators and parent-reported victim-perpetrators had more severe deficit in socio-communication on the Chinese SRS than did the neutrals, whereas no difference in socio-communicative deficit was found between pure perpetrators and the neutrals. The subscale of socio-communication on the Chinese SRS contained items mostly related to social interaction skills, communication skills, and restricted behavior [24]. Deficits in socio-communication may not only increase the conflicts between adolescents with ASD and peers but also decrease the ability of adolescents with ASD to resolve disputes well and build friendships with others. Therefore, the risk of involving in bullying may increase. Two recent studies have found that video modeling [37] and ToM performance training [38] can improve the ability of children and adolescents with ASD to appropriately respond to bullying scenarios and reduce involving in bullying. The other three subscales of autistic social impairment on the Chinese SRS include autism mannerism containing items emphasizing the unique behaviors in ASD (e.g., rigid patterns, plays inappropriately), social awareness reflecting the degree of unawareness in a social context (e.g., cannot recognize something as unfair, laughs inappropriately) and social emotion incorporating the emotional aspects of behaviors (e.g., literally; overly serious facial expressions) [24]. Contrary to our hypothesis, only socio-communication but not autism mannerism, social awareness or social emotion is significantly associated with bullying involvement in adolescents with high functioning ASD. The mechanisms accounting for the various relationships between the four subscales of autistic social impairment on the Chinese SRS and bullying involvement in adolescents with ASD warrant further study.

The present study did not find significant differences in autistic social impairment and ADHD symptoms between victim-perpetrators, pure victims, and pure perpetrators. A previous study on a large sample of adolescents in the community found that victim-perpetrators reported more severe inattention and hyperactivity/impulsivity than pure perpetrators and pure victims [26]. The discrepancy between the results of the present and previous studies indicates that ADHD symptoms have a different role in bullying victimization-perpetration among adolescents with high functioning ASD compared with general adolescent population in the community. However, parent-reported victim-perpetrators had more severe ODD symptoms than did parent-reported pure victims. Owing to bullying victim-perpetrators have the most severe risks of depression, suicide, and alcohol drinking compared with pure victims and pure perpetrators in general adolescent population [26], the possibility of being a bullying victim-perpetrators should be monitored in adolescents with ASD and comorbid a high level of ODD symptoms.

This study has several limitations. First, we did not include teacher-report or peer-report, which may be useful to determine the accuracy of self-report and parent-report on bullying involvement, as well as provide comprehensive information for developing prevention and intervention programs. Second, the study participants were adolescents with high functioning ASD who visited medical units for treatment or survey. Therefore, the results of this study might not be generalizable to all adolescents with high functioning ASD. Adolescents with low functioning ASD or those who did not visit medical units may have the experiences of bullying involvement different from the participants in this study. Third, we examined ADHD and ODD symptoms but not the diagnoses of ADHD and ODD. The diagnoses of ADHD and ODD indicated an increased level of dysfunction resulting from the symptoms. It warrants further study whether comorbid ADHD and ODD increases the risk of bullying involvement compared with subthreshold ADHD and ODD symptoms.

## 5. Implication

Based on the results of the present study, we suggest that mental health and educational professionals must collect multiple sources of information on bullying involvement when intervening bullying involvement of adolescents with ASD. School teachers and parents should establish contacts to communicate their observations on adolescents with ASD with each other. Socio-communication is the main deficit of social impairment related to bullying involvement. Hyperactivity/impulsivity and ODD symptoms were also related to the experiences of bullying involvement. Prevention and intervention programs organized by the school, parents and mental health professionals together may improve the ability of socio-communication and the severity of hyperactivity/impulsivity and ODD symptoms in children and adolescents with ASD to reduce involvement in bullying.

## 6. Conclusions

The present study found that the agreement between self-reported and parent-reported bullying involvement of adolescents with ASD was low. Mental health and educational professionals should collect both the self-reported and parent-reported information to comprehensively detect bullying involvement in adolescents with ASD. The deficit in socio-communication increases the risk of being pure victims and victim-perpetrators. Programs of enhancing socio-communicative ability is needed to help adolescents with ASD reduce the risk of involving in bullying. Parent-reported victim-perpetrators had more severe ODD symptoms than did parent-reported pure victims. The risk of experiencing bullying victimization-perpetration in adolescents with ASD and comorbid a high level of ODD symptoms should be prevented and early intervened.

## Figures and Tables

**Table 1 ijerph-16-01117-t001:** Demographic characteristics, autistic social impairment, and ADHD and ODD symptoms (*n* = 219).

	*n* (%)	Mean (SD)	Range
Sex			
Girls	27 (12.3)		
Boys	192 (87.7)		
Age (years)		13.7 (2.1)	11–18
Marriage status of parents			
Married and living together	189 (86.3)		
Divorced or separated	30 (13.7)		
Paternal education duration (years)		14.7 (2.9)	6–23
Maternal education duration (years)		14.2 (2.5)	6–22
Autistic social impairment on the SRS			
Socio-communication		68.5 (14.0)	31–103
Autism mannerism		33.7 (7.5)	14–52
Social awareness		31.5 (4.6)	18–42
Social emotion		20.7 (3.9)	9–31
ADHD and ODD symptoms on the SNAP-IV			
Inattention		14.7 (6.5)	0–27
Hyperactivity/impulsivity		10.0 (6.7)	0–27
ODD		10.7 (6.3)	0–24

ADHD: attention-deficit/hyperactivity disorder; ODD: oppositional defiant disorder; SNAP-IV: Swanson, Nolan, and Pelham, Version IV Scale; SRS: social responsiveness scale.

**Table 2 ijerph-16-01117-t002:** Self-reported and parent-reported bullying involvement (*n* = 219).

			Parent-Reported			
			Neutral	Pure Victims	Pure Perpetrators	Victim-Perpetrators
			*n* = 91 (41.6%) *n* (%)	*n* = 52 (23.7%) *n* (%)	*n* = 13 (5.9%) *n* (%)	*n* = 63 (28.8%) *n* (%)
Self-reported	Neutral	*n* = 140 (63.9%) *n* (%)	70 (32.0)	31 (14.2)	10 (4.6)	29 (13.2)
	Pure victims	*n* = 39 (17.8%) *n* (%)	12 (5.4)	10 (4.6)	2 (0.9)	15 (6.8)
	Pure perpetrators	*n* = 20 (9.1%) *n* (%)	3 (1.4)	5 (2.3)	1 (0.5)	11 (5.0)
	Victim-perpetrators	*n* = 20 (9.1%) *n* (%)	6 (2.7)	6 (2.7)	0	8 (3.7)

**Table 3 ijerph-16-01117-t003:** The factors related to self-reported bullying involvement: logistic regression ^a^.

	Pure Victims vs. Neutral OR 95% CI of OR		Pure Perpetrators vs. Neutral OR 95% CI of OR		Victim-Perpetrators vs. Neutral OR 95% CI of OR		Victim-Perpetrators vs. Pure Victims OR 95% CI of OR		Victim-Perpetrators vs. Pure Perpetrators OR 95% CI of OR	
Sex	1.209	0.382–3.827	1.244	0.265–5.840	0.553	0.165–1.849	0.457	0.101–2.063	0.444	0.072–2.760
Age	1.132	0.961–1.332	1.077	0.866–1.339	1.006	0.808–1.253	0.879	0.672–1.150	0.924	0.672–1.270
Marriage status of parents	2.151	0.837–5.527	1.471	0.385–5.611	2.083	0.615–7.053	0.969	0.253–3.712	1.417	0.273–7.342
Paternal education duration	0.932	0.823–1.056	1.001	0.849–1.180	0.915	0.776–1.079	0.984	0.827–1.171	0.918	0.742–1.136
Maternal education duration	0.851	0.734–0.986	0.990	0.826–1.188	0.938	0.779–1.129	1.129	0.883–1.444	0.936	0.714–1.227
Socio-communication	1.027	1.000–1.055	1.040	1.003–1.079	1.063	1.020–1.108	1.026	0.984–1.069	1.016	0.968–1.067
Autism mannerism	1.040	0.989–1.092	1.059	0.990–1.133	1.086	1.009–1.168	1.032	0.957–1.114	1.015	0.930–1.108
Social awareness	0.996	0.925–1.073	1.077	0.966–1.201	0.992	0.894–1.100	0.996	0.887–1.119	0.889	0.748–1.057
Social emotion	1.043	0.953–1.140	1.066	0.937–1.212	1.155	1.006–1.327	1.064	0.936–1.210	1.088	0.902–1.313
Inattention	1.001	0.948–1.057	1.090	1.005–1.182	1.033	0.957–1.115	1.025	0.948–1.107	0.947	0.852–1.053
Hyperactivity/impulsivity	1.018	0.962–1.077	1.080	1.010–1.156	1.068	0.999–1.142	1.051	0.972–1.137	0.992	0.919–1.070
ODD	0.998	0.941–1.058	1.054	0.977–1.138	1.018	0.944–1.098	1.017	0.939–1.101	0.974	0.892–1.064

^a^: compared with the neutral group. CI: confidence interval; OR: odds ratio; ODD: oppositional defiant disorder.

**Table 4 ijerph-16-01117-t004:** The factors related to parent-reported bullying involvement: logistic regression ^a^.

	Pure Victims vs. Neutral OR 95% CI of OR		Pure Perpetrators vs. Neutral OR 95% CI of OR		Victim-Perpetrators vs. Neutral OR 95% CI of OR		Victim-Perpetrators vs. Pure victims OR 95% CI of OR		Victim-Perpetrators vs. Pure perpetrators OR 95% CI of OR	
Sex	0.756	0.283–2.019	– ^b^	–	0.945	0.357–2.502	1.250	0.434–3.598	– ^b^	–
Age	1.067	0.910–1.250	1.277	0.975–1.671	1.254	1.066–1.476	1.167	0.970–1.403	0.953	0.697–1.304
Marriage status of parents	1.614	0.550–4.740	3.112	0.708–13.676	2.441	0.935–6.377	1.513	0.548–4.173	0.784	0.187–3.295
Paternal education duration	1.004	0.893–1.128	1.079	0.901–1.291	1.024	0.922–1.136	1.027	0.896–1.177	0.936	0.761–1.152
Maternal education duration	1.036	0.908–1.184	1.029	0.826–1.283	1.008	0.883–1.152	0.968	0.831–1.128	0.974	0.759–1.250
Socio-communication	1.065	1.033–1.098	1.060	1.010–1.111	1.111	1.072–1.153	1.049	1.012–1.086	1.059	0.996–1.125
Autism mannerism	1.122	1.059–1.189	1.109	1.017–1.210	1.193	1.118–1.273	1.069	1.003–1.139	1.077	0.968–1.198
Social awareness	1.075	0.999–1.156	1.092	0.969–1.230	1.107	1.030–1.190	1.038	0.945–1.140	1.000	0.858–1.166
Social emotion	1.137	1.037–1.246	1.234	1.049–1.452	1.230	1.116–1.355	1.088	0.976–1.212	0.975	0.809–1.176
Inattention	1.104	1.039–1.172	1.100	0.996–1.216	1.154	1.088–1.224	1.062	0.998–1.131	1.060	0.963–1.167
Hyperactivity/impulsivity	1.121	1.055–1.190	1.092	1.001–1.191	1.143	1.078–1.212	1.025	0.969–1.084	1.036	0.948–1.133
ODD	1.018	0.957–1.083	1.154	1.036–1.285	1.162	1.090–1.239	1.112	1.045–1.183	1.012	0.923–1.109

^a^: compared with the neutral group; ^b^: no parent-reported pure perpetrator was girl; CI: confidence interval; OR: odds ratio; ODD: oppositional defiant disorder.

**Table 5 ijerph-16-01117-t005:** The factors related to self-reported and parent-reported bullying involvement: logistic regression ^a^.

	Self-Reported						Parent-Reported							
	Pure Victims vs. Neutral aOR 95% CI of OR		Pure Perpetrators vs. Neutral aOR 95% CI of OR		Victim-Perpetrators vs. Neutral aOR 95% CI of OR		Pure Victims vs. Neutral aOR 95% CI of OR		Pure Perpetrators vs. Neutral aOR 95% CI of OR		Victim-Perpetrators vs. Neutral aOR 95% CI of OR		Victim-Perpetrators vs. Pure Victims aOR 95% CI of OR	
Age											1.339	1.073–1.671		
Maternal education	0.845	0.727–0.983												
Socio-communication	1.028	1.001–1.056	1.021	0.972–1.072	1.101	1.011–1.198	1.051	0.976–1.132	0.999	0.891–1.121	1.121	1.037–1.213	1.035	0.970–1.106
Autism mannerism					0.927	0.788–1.092	1.079	0.952–1.223	1.031	0.827–1.284	1.077	0.925–1.254	0.982	0.872–1.105
Social awareness											0.877	0.771–1.001		
Social emotion					1.004	0.804–1.252	0.881	0.739–1.051	1.190	0.874–1.621	0.881	0.741–1.048		
Inattention			1.024	0.908–1.156			0.971	0.885–1.067			1.012	0.915–1.119		
Hyperactivity/impulsivity			1.043	0.951–1.143			1.084	1.000–1.175	0.967	0.836–1.119	0.982	0.888–1.087		
ODD									1.162	0.995–1.356	1.127	1.022–1.243	1.091	1.020–1.1666

^a^: compared with the neutral group; aOR: adjusted odds ratio; CI: confidence interval; ODD: oppositional defiant disorder.

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
