# Peer review of "Self-Reported and Parent-Reported School Bullying in Adolescents with High Functioning Autism Spectrum Disorder: The Roles of Autistic Social Impairment, Attention-Deficit/Hyperactivity and Oppositional Defiant Disorder Symptoms"

_ijerph, 2019, doi:10.3390/ijerph16071117_

Round 1
Reviewer 1 Report
The manuscript titled ‘Self-Reported and Parent-Reported School Bullying in Adolescents with High Functioning Autism Spectrum Disorder: The Roles of autistic Social Impairment, Attention-Deficit/Hyperactivity and Oppositional Defiant Disorder Symptoms’ by Wen-Jiun Chou et. al., examined the prevalence of self-reported and parent-reported bullying victimization, perpetration, and victimization-perpetration and the association of demographic characteristics and the level of autistic social impairment and ADHD and ODD symptoms among with high functioning ASD. Chinese version of the School Bullying Experience Questionnaire were used to evaluate the self-reported and parent-reported bullying victimization, perpetration, and victimization-perpetration. Total population of 219 were from Taiwan following some specific criteria. The study period were from August 2013 to July 2016. Here parents reported higher rates of adolescents' pure bullying victimization and victimization-perpetration than that of self reported, whereas the rate of pure bullying perpetration was lower. it is also noticed that parents identified 50% of self-reported neutrals as pure victims, pure perpetrators, or victims-perpetrators. Only 30% of the self-reported victim-perpetrators was reported by the parents as the neutrals. In this study, parent-reported victim perpetrators had more severe ODD symptoms than did parent-reported pure victims.
In summary, this is an interesting paper using Chinese version of questionnaire to suggest low agreement between self-reported and parent-reported bullying involvement of adolescent with ASD. The manuscript has a number of strengths, but also limitations such as lake of mental health and educational professionals. Nevertheless, I think it is of sufficient importance to be given consideration. I recommend for the publication of this research findings.
Author Response
Thank you for your comment. We did not include teacher-report or peer-report to determine the accuracy of self-report and parent-report on bullying involvement. We listed it as one of limitations of this study.
Reviewer 2 Report
The study examined parent-reported bullying victimization, perpetration, and victim-perpetration and their associations of ADHD and ODD symptoms. The sample included 219 adolescents with ASD. The introduction is clear and effective. The sources are relevant to building your argument and you do well in discussing the prevalence of the issue as well. The hypotheses are grouped up clearly as well. The findings were also well-discussed. For the most part, I believe your study is valuable and it contributes to our understanding of adolescents with ADHD/ODD symptoms and their bullying behaviors. I also appreciated the addition of the social communication deficits. However, I would like to see literature about social communication deficits in particular in your literature review/intro section. Please include a couple of sources and intertwine it into your literature review.
In the methods section you indicate 219 adolescents and their parents, does this mean 219 adolescents and 219 parents? OR, more parents? Can you mention more about the parents? Was it their father, mother, or whom? Did you test to see if the parents also suffer from ADHD or ODD? How might the results be different if they also had it versus not? Perhaps discuss that in the discussion section.
As far as the questionnaire, how long did it take on average to complete? Was it paper-based? Talk a little more about this.
The measures selected are valid and reliable. You did well in providing samples and also providing the alpha reliabilities for each measure.
The analysis and tables are clear and effective. I appreciate the level of depth in explaining the analysis and the results.
The sample size is rather modest. Can you include your power analysis to ensure your sample size is appropriate for the rigorous analyses that you conducted?
In the discussion you mention that the “agreement between self-reported and parent-reported bullying involvement of adolescents with ASD was low” – however, I think you need to discuss why this might be the case. What are some reasons for this finding? What do you think?
As far as the limitations section, it appears quite vague. Provide another sentence for each limitation. For instance, what benefit would adding teacher-report/peer-reports do to future studies?
Consider adding a short implications section. What are the implications for this ADHD/ODD population of adolescents and their parents? What value do your findings add to the community? What are the implications for parents and teachers?
Overall, the paper is well-written and it does provide rigorous analyses. I believe the paper has potential.
Author Response
Comment 1
I would like to see literature about social communication deficits in particular in your literature review/intro section. Please include a couple of sources and intertwine it into your literature review.
Response
We appreciate your suggestion. We added a paragraph as below to review the possible etiologies accounting for the relationship between social communication deficits and bullying involving among the individuals with ASD in Introduction section. Please refer to line 84-94.
“Social communication deficits may be the results of deficits in theory of mind (ToM) performance among the individuals with ASD [18, 19]. The individuals with deficits in ToM performance may have difficulties in understanding others’ mental states and in predicting others’ behaviors resulted from their mental states [20]. These difficulties may compromise social interactions of ASD children and adolescents with peers, make them look odd, and increase the chance to be the targets of bullying. Social communication deficits can make the individual with ASD misinterpret others’ intention and have inappropriate reactions, which may be interpreted by their peers as provocative actions and become the flash point of bullying [21]. Social communication deficits may also limit the ability of the individuals with ASD to build up friendship with others and therefore reduce the possibility to receive protection and assistance from others when they are bullied.”
Comment 2
In the methods section you indicate 219 adolescents and their parents, does this mean 219 adolescents and 219 parents? OR, more parents? Can you mention more about the parents? Was it their father, mother, or whom? Did you test to see if the parents also suffer from ADHD or ODD? How might the results be different if they also had it versus not? Perhaps discuss that in the discussion section.
Response
1. We revised the sentences in Methods section as below. Please refer to line 143-145.
“A total of 228 adolescent-parent dyads were invited into this study. Of them, 219 (96.1%) adolescent-parent dyads (219 adolescents with high-functioning ASD, 175 mothers, 33 fathers, and 11 other relatives) agreed to participate in this study…”
2. We did not evaluate the mental illnesses of the parents. We added the paragraph in Discussion section to discuss the influence of parental mental health status on reporting adolescents’ bullying involvement as below. Please refer to line 302-309.
“…although the present study did not assess parental mental health statuses, parents’ psychiatric disorders may influence their observation and report on their offspring’s experiences of bullying involvement. For example, depression may reduce parents’ ability to detect what their offspring encounter in school. Research found that mother’s depression even increased the risk of bullying in their sons [36]. It is also possible that parental ASD, ADHD and ODD may influence parents’ interpretation of social interaction in adolescents. For example, impulsivity may limit the ability of the parents with ASD, ADHD or ODD to comprehensively understand adolescents’ social conflicts and attribute them as bullying or not bullying involvement.”
Comment 3
As far as the questionnaire, how long did it take on average to complete? Was it paper-based? Talk a little more about this.
Response
Research assistants conducted the interview to collect adolescents’ self-reported school bullying experiences based on the C-SBEQ. It took five minutes on average for the adolescents to complete the interview. The parents spent 30 minutes on average to complete the pencil-and-paper C-SBEQ, the Chinese SRS, short version of SNAP-IV and the questionnaire for demographic characteristics. We added the description in line 194-198.
Comment 4
The measures selected are valid and reliable. You did well in providing samples and also providing the alpha reliabilities for each measure.
Response
Thank you for your comment.
Comment 5
The analysis and tables are clear and effective. I appreciate the level of depth in explaining the analysis and the results.
Response
Thank you for your comment.
Comment 6
The sample size is rather modest. Can you include your power analysis to ensure your sample size is appropriate for the rigorous analyses that you conducted?
Response
We included our power analysis for determining sample size as below in Methods section. Please refer to line 148-154.
“The result of the systematic review study found that over 50% of children and adolescents with ASD involved in bullying [2]. The present study focused on adolescents with high functioning ASD, and therefore the prevalence of bullying involvement was hypothesized to be 35-40%. Based on a small effect size in logistic regression (odds ratio: 1.0-1.5) with an alpha level of .05, a power of .80, and the hypothesized rate of bullying involvement 35-40%, 200 participants were deemed to be sufficient to test the hypothesis [30]. The sample of 219 participants was thus determined as adequate.”
Comment 7
In the discussion you mention that the “agreement between self-reported and parent-reported bullying involvement of adolescents with ASD was low” – however, I think you need to discuss why this might be the case. What are some reasons for this finding? What do you think?
Response
We added discussion on the possible etiologies accounting for the low agreement between self-reported and parent-reported bullying involvement of adolescents with ASD and the implication of the results as below into Discussion section of the revised manuscript. Please refer to line 278-290.
“Adolescents may not agree with the perspective of adults on whether some behaviors should be regarded as bullying [35]. ASD may aggravate disagreement of bullying involvement between adolescents and parents. Adolescents with ASD may misinterpret bullying situations as non-bullying [13,14]. Furthermore, the classmates and teachers of adolescents with ASD may interpret adolescents’ autistic behaviors at school as bullying behaviors and report them to their parents, whereas the adolescents with ASD may deny that they have intent to bully others. Adolescents with ASD and their families may also experience difficulties in interacting and communicating with each other. Such difficulties may further hinder parents from detecting adolescents’ involvement in bullying. Given the crucial role of parents in the prevention of and intervention for bullying involvement, the results of the present study indicate that enhancing parents’ knowledge of ASD adolescents’ bullying involvement at school is an essential step to address bullying. Moreover, mental health and educational professionals must take the self-reported and parent-reported bullying involvement into consideration simultaneously and should not rely on sole information when intervening bullying involvement of adolescents with ASD.”
Comment 8
As far as the limitations section, it appears quite vague. Provide another sentence for each limitation. For instance, what benefit would adding teacher-report/peer-reports do to future studies?
Response
We revised the limitations section as below. Please refer to line 342-352.
“First, we did not include teacher-report or peer-report, which may be useful to determine the accuracy of self-report and parent-report on bullying involvement, as well as provide comprehensive information for developing prevention and intervention programs. Second, the study participants were adolescents with high functioning ASD who visited medical units for treatment or survey. Therefore, the results of this study might not be generalizable to all adolescents with high functioning ASD. Adolescents with low functioning ASD or did not visit medical units may have the experiences of bullying involvement different from the participants in this study. Third, we examined ADHD and ODD symptoms but not the diagnoses of ADHD and ODD. The diagnoses of ADHD and ODD indicated an increased level of dysfunction resulted from the symptoms. It warrants further study whether comorbid ADHD and ODD increases the risk of bullying involvement compared with subthreshold ADHD and ODD symptoms.”
Comment 9
Consider adding a short implications section. What are the implications for this ADHD/ODD population of adolescents and their parents? What value do your findings add to the community? What are the implications for parents and teachers?
Response
We added an implication section as below to the revised manuscript. Please refer to line 353-362.
“Based on the results of the present study, we suggested that mental health and educational professionals must collect multiple sources of information on bullying involvement when intervening bullying involvement of adolescents with ASD. School teachers and parents should establish contacts to communicate their observation on adolescents with ASD with each other. Socio-communication is the main deficit of social impairment related to bullying involvement. Hyperactivity/impulsivity and ODD symptoms were also related to the experiences of bullying involvement. Prevention and intervention programs organized by the school, parents and mental health professionals together may improve the ability of socio-communication and the severities of hyperactivity/impulsivity and ODD symptoms in children and adolescents with ASD to reduce involving in bullying.”
Comment 10
Overall, the paper is well-written and it does provide rigorous analyses. I believe the paper has potential.
Response
Thank you for your comment.
Reviewer 3 Report
Thank you for the opportunity to review this manuscript. While
the research study focuses on an interesting important topic, there were
areas where the manuscript could have been improved, considerably
before resubmission. As a result, I have noted
some of these issues below.
Abstract section is way too complicated and confusing, especially lines 26-36. This needs to be rewritten by authors.
Methods section: For the reader, the methods need to be explained further, especially key research objectives could be justified more strongly and clearly.
Discussion and Conclusion section: in this paper discussion is more like a results/conclusion section. The discussion section should link the findings (which are very interesting) back to the greater body of literature. This section must be reviewed very carefully.
Conflict of interest statement is missing and particularly important with funded research.
Author Response
Comment 1
Abstract section is way too complicated and confusing, especially lines 26-36. This needs to be rewritten by authors.
Response
Thank you for your suggestion. We rewrote the content of Abstract section to make it simpler and clearer. Please refer to line 28-42.
Comment 2
Methods section: For the reader, the methods need to be explained further, especially key research objectives could be justified more strongly and clearly.
Response
Thank you for your suggestion. In the revised manuscript we revised the contents of 2.1. Participants (line 143-145 and 148-154) and 2.3. Procedures (line 194-198) to explain more about the research objectives and procedures to collect data.
Comment 3
Discussion and Conclusion section: in this paper discussion is more like a results/conclusion section. The discussion section should link the findings (which are very interesting) back to the greater body of literature. This section must be reviewed very carefully.
Response
Thank you for your suggestion. In the revised manuscript we rewrote the first paragraph to describe the possible etiologies accounting for low agreement between self-reported and parent-reported bullying involvement of adolescents with ASD and the implication (line 278-290). We added the content of the second paragraph to discuss the role of parental mental health statuses for parent-reported adolescents’ bullying involvement (line 302-309). We also added discussion on the relationship of deficits in socio-communication with bullying involvement in the third paragraph (line 316-319). The content of paragraph describing the limitations of the present study was revised thoroughly (line 342-352). We also added an Implication section into Discussion section (line 353-362).
Comment 4
Conflict of interest statement is missing and particularly important with funded research.
Response
Thank you for your reminding. We added Conflict of interest statement into the revised manuscript. Please refer to line 385-387.